# Decoding the Mechanism of Action of Rapid-Acting Antidepressant Treatment Strategies: Does Gender Matter?

**DOI:** 10.3390/ijms20040949

**Published:** 2019-02-22

**Authors:** David P. Herzog, Gregers Wegener, Klaus Lieb, Marianne B. Müller, Giulia Treccani

**Affiliations:** 1Department of Psychiatry and Psychotherapy, Johannes Gutenberg University Medical Center Mainz, Untere Zahlbacher Straße 8, 55131 Mainz, Germany; daherzog@uni-mainz.de (D.P.H.); klaus.lieb@unimedizin-mainz.de (K.L.); 2Focus Program Translational Neurosciences, Johannes Gutenberg University Medical Center Mainz, Langenbeckstraße 1, 55131 Mainz, Germany; 3Translational Neuropsychiatry Unit, Department of Clinical Medicine, Aarhus University, Skovagervej 2, 8240 Risskov, Denmark; wegener@clin.au.dk

**Keywords:** gender, sex difference, depression, antidepressant, rapid-acting, antidepressant, Ketamine, endocrinology, (2R,6R)-Hydroxynorketamine, electroconvulsive therapy

## Abstract

Gender differences play a pivotal role in the pathophysiology and treatment of major depressive disorder. This is strongly supported by a mean 2:1 female-male ratio of depression consistently observed throughout studies in developed nations. Considering the urgent need to tailor individualized treatment strategies to fight depression more efficiently, a more precise understanding of gender-specific aspects in the pathophysiology and treatment of depressive disorders is fundamental. However, current treatment guidelines almost entirely neglect gender as a potentially relevant factor. Similarly, the vast majority of animal experiments analysing antidepressant treatment in rodent models exclusively uses male animals and does not consider gender-specific effects. Based on the growing interest in innovative and rapid-acting treatment approaches in depression, such as the administration of ketamine, its metabolites or electroconvulsive therapy, this review article summarizes the evidence supporting the importance of gender in modulating response to rapid acting antidepressant treatment. We provide an overview on the current state of knowledge and propose a framework for rodent experiments to ultimately decode gender-dependent differences in molecular and behavioural mechanisms involved in shaping treatment response.

## 1. Introduction

### 1.1. Does Gender Matter in Major Depressive Disorder (MDD)?

MDD poses a serious threat to global mental health. It is the second leading cause of disability worldwide [1], with more than 300 million people affected [2]. The lifetime prevalence for MDD is 2–3 times higher in women compared to men [3], however current diagnosis and treatment guidelines for MDD around the world (e.g., [4]) do not really consider gender differences. Apart from general recommendations regarding sex hormone dysfunction, premenstrual and menopausal hormonal changes that can be causal factors contributing to the development of depression-like syndromes, the diagnosis of MDD relies on still unisex criteria (ICD-10 or DSM-5). Interestingly, the incidence of new onset of depression cases drops tremendously after menopause [5]. These women are susceptible to the same extent to MDD as men, thus highlighting a crucial role of sexual hormones in the pathophysiological mechanisms of depression.

Oestrogens influence synaptic plasticity, neurotransmission, neurodegeneration and cognitive function [6]. In addition, several studies revealed a neuroprotective role of progesterone in rodent and in humans [7]. Therefore, the underlying pathways driven by hormones and other molecular players are in the spotlight of gender-specific MDD research, with the intent of possibly bridging the gap between gender-specific disease parameters (i.e., higher symptom severity [8,9,10], an earlier onset of disease [9,10] and an increased duration of depressive episodes [9,10]) and underlying neurobiological and molecular changes.

### 1.2. Genetics, Epigenetics and Hormones: Powerful Players Shaping Gender-Specificity of MDD

The sexually dimorphic anatomy and function of the brain is strongly influenced by a broad variety of parameters, such as hormonal status, variance in body fat, liver metabolism, the transcription machinery, gene accessibility via epigenetic modifications and others [11,12,13,14]. Moreover, the heritability of MDD is higher in women than in men implicating an increased genetic vulnerability [15].

A longitudinal study by Bundy et al. showed that 198 genes were differentially expressed in the hippocampus between male and female mice across developmental stages [16]. The older the animals were, the more differentially expressed genes were found between male and female mice, indicating the importance and the increase over time in sexual dimorphism of the genome. Indeed, the difference of the transcriptome profile is bigger between female rodents with high and low hormonal states than between female and male rodents [17]. A recent translational study from the Nestler group focused on sex differences of transcriptome profiles comparing gender in both humans and mice [18]. In both species, the authors revealed that there is only a limited overlap of regulated genes between males and females in several brain regions [18]. These data highlighted that the transcriptome profiles are gender-specific both in depressed patients and in animal models resembling some of the features of depression. In depressed patients, only 5–10% of genes were shared between women and men [18], while in the animals were 20% [18]. Finally, by means of a translational approach, the researchers were able to identify and validate gender-specific candidate genes for depression, namely *Dusp6* for female and *Emx1* for male gender [18].

In the complex symphony of genetic risk load and environmental factors of depression, epigenetics plays a pivotal role in the gene-environmental interaction [19]. Epigenetics (i.e., all mechanisms to modify chromatin accessibility) affects brain functions as well as disease conditions [20]. In 2015, Nugent and colleagues reported that in male rodents the enzyme DNA methyltransferase 3a was required to keep a male phenotype [21]. Manipulation of this process led to feminization of the brain, thus displaying the importance of epigenetic mechanisms for gender determination of the organism. A recent review by Marija Kundakovic highlighted the need to include both sexes in research studies of depression and at the same time to consider the hormonal state of female to study brain function in healthy and disease conditions [11].

The most relevant risk factor of MDD is stress. Stress lowers the threshold for an organism to develop mental diseases like MDD. Stress can be similarly used in animal models to study mental illnesses, as it is a highly conserved and evolutionary important factor across species. Response to stress is highly gender-specific. Female rodents have a greater sensitivity to stress with a more prominent stress response, a longer period of recovery and a more active hypothalamic–pituitary–adrenal (HPA) axis [22,23]. During the stress response, the induction of immediate early gene *Fos* after acute stress is greater in female rodents [24], especially in the hippocampus, where immediate early gene expression is reported to be more pronounced [25]. At the morphological level, stress has different impact on gender. Indeed, chronic stress reduced dendritic complexity and spine remodelling in hippocampal pyramidal cells of male rodents but not in females [26,27]. Moreover, social deprivation experiments using single housing of rodents over a long period of time revealed more anxious and anhedonia-like phenotypes in both male and female rodents, although the effect was more pronounced in males [28]. For a more in-depth and recent review of gender specificity of behavioural tests in terms of depression, depression treatment and stress see [12].

The discovery of the HPA axis together with the detection of glucocorticoid receptors found in hypothalamus and hippocampus [29], led to the conclusion that hormones and the brain were strongly connected and influenced by each other. This is supported by a plethora of interesting findings (Figure 1): Hormonal alterations during the menstrual cycle affect females across species. Elevated oestrogen levels mediate an enhanced hippocampal spine density [30] and plasticity [31]. Hagemann et al. revealed that during ovulation women had a significant increase in grey matter volume [32]. Finally, the end of the menstrual cycle and by that, the decline of sexual hormones oestrogen and progesterone may induce depression-like symptoms [33]. Sex differences in the stress-induced remodelling of dendrites and synapses in brain regions such as the hippocampus or the prefrontal cortex first emerge in puberty [34], thus highlighting the link between female susceptibility to stress related disorders and hormone effects, mainly oestrogen and progesterone [14]. For a review article summarizing the hormonal impact on the female brain see [13].

To support the evidence of the importance of a sexual dimorphic brain, several studies revealed a crosstalk between sexual hormones and key neurotransmitters altered in depression. Amin and colleagues showed that in rodents, brain serotonin pharmacodynamics and –kinetics are affected by oestrogens [35]. Serotonin, the major player in the monoaminergic hypothesis depression, is also influenced by gender; human studies revealed that women had higher levels of serotonin, serotonin metabolites [36] and serotonin transporters [37]. This might also explain why a lack of serotonin levels has a stronger effect in women. Finally, brain-derived neurotrophic factor (BDNF), a key mediator of neurogenesis and neuronal survival altered in depression and inversely associated with depressive symptoms [38] expression, synthesis and function is influenced by oestradiol, whereas BDNF itself is a downstream mediator activated by oestradiol signalling in the hippocampus [39].

### 1.3. Gender-Specific Differences in MDD Therapy

In line with the lack of gender-specific recommendations for the diagnosis of MDD, there is also a neglect of gender in depression therapy. During antidepressant treatment, women generally show higher response rates than men [40]. What is the evidence for gender-specificity of antidepressant treatment? A meta-analysis covering 30 randomized controlled trials found no gender-specificity of tricyclic antidepressant drugs (TCAs) like imipramine [41], whereas other studies reported a superior response rate of males receiving TCAs (e.g., [42,43]). It was also shown that females after menopause benefit from TCA treatment compared to pre-menopausal women, similarly like males [42,43]. Selective serotonin reuptake inhibitors (SSRI) like fluoxetine or citalopram, commonly used antidepressant agents, induced a response only in 50% of SSRI-treated patients while 70% of the same patients do not have full remission after 12 weeks of treatment with SSRIs [44]. There are numerous studies reporting that SSRIs have a higher efficacy in women [42,43,45,46]. In addition, SSRI and hormone replacement therapy was reported to be beneficial for women after menopause [47,48]. In contrast, a meta-analysis from Cuijpers et al. did not detect sex differences in the treatment with either SSRIs or TCAs [49]. The inconsistencies within those reports are not easy to be explained. One reason might be highly different inclusion and exclusion criteria of clinical trials. These might lead to different patient stratification, thus creating a huge bias and complicating the interpretation of the results. Another possible reason might be that psychiatric nosology and diagnosing is a highly artificial process. Two persons diagnosed with depression may express completely different symptoms, however in clinical trials, they might be put together as namely “depressed patient.” A recent National Institute of Mental Health (NIMH) initiative called the Research Domain Criteria (RDoC), introduces a transdiagnostic approach suited to specify symptoms and phenotypes at individual level [50].

For a detailed review about the evidence of gender-specificity of *classical* antidepressant drug treatment see [14].

Additional factors have been reported to play a role in gender differences in antidepressant efficacy. Different pharmacokinetics of TCAs [51,52] could explain why women report more drug related side effects and why women seem to prefer medication with SSRIs. Women showed a superior adherence to continuous antidepressant drug treatment [53].

### 1.4. Why We Need Antidepressant Treatment Approaches with a Rapid Onset of Action

After initiating antidepressant drug treatment with classical antidepressant compounds (i.e., TCAs or SSRIs) there is a latency lasting up to four weeks before a detectable treatment response becomes evident. A possible loss of antidepressant treatment efficacy in long term-treated patients, the danger of manic switches and several prominent side effects are further limitations of antidepressant treatment [54,55]. In addition, the large heterogeneity of antidepressant treatment response and the lack of biomarkers to monitor or stratify disease state, to facilitate diagnostic decisions or to predict treatment success already early turn antidepressant drug treatment into a large “trial and error” game [56]. To overcome the current limitations in neuropsychopharmacology, increasing attention is given to biomarker research by using truly translational research projects, combining animal and human cohorts in the same study design. The accessibility of the central nervous system for the validation of potent drug candidate structures and the possibility of biomarker studies in human and animal blood offer a unique and meaningful platform to really contribute to the development of novel antidepressant agents and improve MDD therapy [57]. Another promising approach is the emergence of rapid-acting antidepressant treatments, which may possess a distinct mechanism of action than the conventional antidepressants, promote faster recovery and thereby overcome the high socio-economic costs of long-lasting depression courses. Among those, electroconvulsive therapy, ketamine and other compounds showing similar properties, cause higher response rate also in treatment resistant, heavily depressed patients - for some patients even within hours. In the following paragraphs we will highlight the major findings of this class of therapy and discuss the implication of gender in their mechanism of action.

## 2. Gender-Specific Differences in the Molecular Mechanisms of Rapid-Acting Antidepressant Drugs

### 2.1. Ketamine: Evidence of Gender-Specific Differences in the Effect on the Brain

Recently, the discovery of drugs with rapid acting antidepressant efficacy has built the basis for novel treatment strategies in MDD. Among those, ketamine, a non-competitive NMDA receptor antagonist, originally introduced as a dissociative anaesthetic, revealed to induce a rapid (within hours) and sustained (up to 1 week) antidepressant effect in treatment-resistant [58,59]. Furthermore, its rapid action on suicide ideation significantly improves the management of acute MDD treatment [60,61].

Ketamine has been shown to exert antidepressant-like effects in different animal models of depression by activation of the mammalian target of rapamycin (mTOR), by increasing the expression of synaptic proteins and increasing synaptogenesis in the prefrontal cortex [62]. The mechanism by which ketamine produces fast antidepressant-like effects depends on the rapid synthesis of brain-derived neurotrophic factor [63] and on dendritic release/translation of BDNF. 

Preliminary clinical observations suggest gender differences to play a role in ketamine metabolism and clearance in a dose dependent manner. However, only recently a more comprehensive characterization of the gender differences in the effects of ketamine has been done. Female C57BL/6J stress-naïve mice were more sensitive to the rapid and sustained antidepressant-like effects of ketamine in the forced swim test (FST) and responsive to lower doses of ketamine [64]. Moreover, the female mice responded earlier to a single injection of ketamine than the male mice.

A possible explanation may lie in sex differences of ketamine pharmacokinetics. Female rats exhibited greater concentrations of ketamine and norketamine over the first 30 min following treatment in both brain and plasma, due to slower clearance rates and longer half-lives. Gender differences can influence the metabolism of ketamine and therefore the amount of ketamine and norketamine reaching brain areas [65].

In another study, instead, oestrogens augmented the effect of ketamine and its metabolites (2R,6R)-HNK and (2S,6S)-HNK via induction of the CYP2A6 and CYP2B6 enzymes responsible for ketamine’s biotransformation into its active metabolites [66]. 

The pharmacodynamics of ketamine is also modified in a gender specific way with a possible synergism with sexual hormone. Female enhanced sensitivity to ketamine during proestrus was mediated via oestradiol activation of oestrogen receptor (ER)alpha and ERbeta, leading to greater activation of synaptic plasticity related kinases within prefrontal cortex and hippocampus (Table 1) [67]. 

Different molecular pathways are activated by ketamine in a gender specific manner. Indeed male chronic isolated rats showed an increased spine density after ketamine treatment in medial prefrontal cortex via restoration of synapsin 1, PSD95 and GluR1 levels while those proteins were not altered in female rats after ketamine treatment [28]. 

However, the literature reported as well studies showing no effects of gender in the effects of ketamine. Indeed, a recent work revealed no effects of gender in the acute and chronic effects of ketamine in ICR mice [68].

### 2.2. Rapid Antidepressant-like Effects with Less Side Effects? Emerging Data on Ketamine Metabolites

Recently, the major ketamine metabolites have been drawn into focus of international research studies. In 2016, Zanos and colleagues showed that the ketamine metabolite (2R,6R)-hydroxynorketamine (HNK) produced antidepressant-like effects in mice similarly to those of ketamine but without the ketamine-associated side effects [69]. They were the first to propose a NMDA receptor-independent mechanism for the antidepressant-like effects of ketamine and its metabolites, indicating an important role of AMPA receptors [69]. Although the relevance of HNK and the question of AMPA receptor involvement still remains under debate ([70,71] and Table 2), HNK is a very interesting compound offering the great potential of rapid antidepressant-like effects without strong side effects.

Chou and colleagues applied the learned helplessness paradigm in male and female rats as a depression model in rodents [72]. They could show that HNK rapidly rescued depression-like conditions assessed by using the FST and the sucrose preference test (SPT) [72]. In addition, they did not detect a difference between female and male rats with respect to HNK effects and no effect at all using (2S,6S)-hydroxynorketamine [72]. Yamaguchi and colleagues recently reported that in male mice using the lipopolysaccharide (LPS) inflammation model of depression the metabolism from R-ketamine to HNK is not exclusively essential for the antidepressant-like effects of ketamine in rodents [70]. They validated this finding using cytochrome p 450 inhibitors. By combining a microdialysis experiment with the FST, HNK was found both in plasma and in brain as an antidepressant-like acting metabolite [73]. At the morphological level, HNK produces an increase of structural plasticity in murine and human dopaminergic neurons [74]. Similarly, Collo et al. reported that in human dopaminergic neurons HNK produced effects on dendritic outgrowth similar to those seen with ketamine [75] and Yao et al. showed that ketamine and HNK effects on AMPA-receptors and synapse alterations in the murine mesolimbic system are strongly aligned [76].

On the other hand, some publications show a lack of response to HNK in rodents. Shirayama et al. reported that they could not find an effect of HNK in a rat model of learned helplessness [77] using the Conditioned Avoidance Test. On top of that, Yang and colleagues were only able to find a very weak effect of HNK in an LPS model of depression in mice and did not find an effect in a chronic social defeat model of depression [78] using the FST, the Tail Suspension Test and the SPT.

To our knowledge, there is only one paper [72] focusing on the gender aspect of HNK treatment. In this study, no difference of the antidepressant-like effects of HNK could be observed. It is important to fill this gap, especially because a more pronounced effect of ketamine in females has been repeatedly reported both in rodents and humans.

### 2.3. Other Rapid-Acting Antidepressant Agents

Rapid-acting antidepressant agents share some key neurobiological pathways, which probably mediate their antidepressant-like effects [79]. By altering glutamate transmission, they enhance mTOR signalling, which leads to increased BDNF levels, a process strongly connected to enhanced synaptic activity and plasticity in the prefrontal cortex (PFC). Besides ketamine and its metabolites, there are several substances and molecules known, which fit into this mechanistic framework and at the same time have shown antidepressant-like effects (Table 3). For recent in-depth reviews see [71,80].

In line with the glutamate hypothesis of depression and depression treatment, pharmacological research focuses on the role of NMDA and AMPA receptors. Based on the evidence showing that the NMDA receptor antagonist ketamine produced sustained antidepressant effects in both humans and rodents, other ways to manipulate glutamate transmission were subsequently examined. Scopolamine, a muscarinic acetylcholine receptor (mAchR) antagonist, produced rapid, antidepressant effects even in treatment resistant depressed patients [81]. Rodent studies revealed that type 1 and 2 mAchRs are exclusively responsible for these effects [82], which led to enhanced AMPA receptor signalling and mTOR activation.

GLYX-13 which is also known as Rapastinel—targets the outer surface of the NMDA receptor without occupying its ion pore and acts as a partial agonist [71,80]. That is probably why a single injection of GLYX-13 can decrease depression scores in patients [84], without inducing ketamine-related side effects. The antidepressant effect appeared after two hours and persisted for at least 7 days. This compound was effective in phase 2 clinical trials [83]. In addition, the direct and specific manipulation of type 2 and 3 metabotropic glutamate receptors emerged as an interesting approach. By enhancing AMPA receptor signalling, antagonists of these receptors produced antidepressant-like effects in rodents, without ketamine-related side effects [85,86,87]. GABA_A_ receptor signalling reduces glutamate transmission, which explains why GABA receptor antagonists and modulators are also important in the search for rapid-acting antidepressant drugs. Negative allosteric modulation of the alpha 5 subunit of the type A GABA receptor provides cortex- and hippocampus specific [95] antidepressant-like effects [88] in an animal study by Fischell et al. These drugs have also been shown to lack the ketamine-related side effects [89].

Cannabidiol (CBD) is the non-psychotomimetic compound of *Cannabis sativa*. Anxiolytic [96] and antidepressant-like effects [97] have been reported in the literature. More recently, it was shown that type 1A serotonin receptor (5-HT_1A_) is important for the effects of cannabidiol. In a study with rats [91], cannabidiol was injected bilaterally into ventromedial parts of the PFC. Acute antidepressant-like effects were observed, which were dependent on 5-HT_1A_ and cannabinoid receptor CB1 signalling. Sales et al. have shown that i.p. injection of cannabidiol also led to an acute (30 min) antidepressant response in rodents [90]. They were able to observe an increase in synaptic plasticity in the PFC, accompanied by elevated BDNF levels in PFC and hippocampus. The antidepressant-like effects could be prevented by blocking the mTOR pathway. Two studies with CBD found that CBD in the saccharin preference test showed a pro-hedonic effect of CBD in male and female Wistar Kyoto rats (WKY) and decreased immobility in the FST in male Flinders Sensitive Line rats (FSL) and male and female WKY but not female FSL [92,93].

Inducing structural and functional plasticity in the PFC is a key process of antidepressant action. It has been shown that psychedelics such as lysergic acid diethylamide (LSD) and others (see Table 3) are also able to exert rapid antidepressant-like effects in humans and rodents. Recently, Ly et al. showed that LSD and alike agents produced a robust increase in neuritogenesis and spinogenesis in vitro and in vivo [94]. They were able to show these effects spanning a range from drosophila larvae, zebrafish embryos and rats, leading to the conclusion that it is a process highly conserved during evolution. Finally, they could show that mTOR and type 2A serotonin receptor play crucial roles in inducing these morphological and molecular changes.

## 3. Non-Pharmacological, Rapid-Acting Treatment for MDD: Electroconvulsive Therapy

### 3.1. Molecular Pathways Shaping the Effect of Stimulating the Brain

Brain stimulation techniques represent an important part of last-line treatment options for MDD therapy. They are usually considered for patients with severe depression and treatment-resistant MDD courses. By applying electricity, all brain stimulation approaches interfere with activation and/or inactivation of certain brain circuits, brain regions and molecular pathways. Electroconvulsive therapy (ECT) is one of the oldest treatment paradigms in psychiatry, which is effective and approved for treatment-resistant depression, depression with psychotic symptoms, mania, catatonia and treatment-resistant schizophrenia [98]. Under anaesthesia and muscle relaxation, it induces a generalized seizure [99,100]. Some patients report an improved depressive symptomatology score already after their first ECT session within hours after 2-3 sessions reliable and sustained antidepressant effects occur even in chronic, severely ill patients and after 8 bilateral sessions the average patient shows full remission of symptoms [101].

Neuroinflammation, reduced levels of monoaminergic neurotransmitters and altered HPA axis activity have been identified as key mechanisms in depression pathophysiology [102]. Similar to pharmacological treatment with antidepressant drugs [103], long-term ECT reduces the activation of the immune system [104]. Specifically, Yrondi et al. described an immediate immuno-inflammatory response after the first ECT session, which was reversed after long-term ECT [105]. By normalizing the brain immuno-inflammatory state with ECT, biomarkers linked to antidepressant-like effects—like serum BDNF—are upregulated [106]. ECT also has a strong effect on modulating the HPA axis activity, with the neuroendocrine system being closely linked to depression and antidepressant response. By means of neuroendocrine challenge tests (dexamethasone suppression test and the combined dexamethasone/corticotropin releasing hormone test), a reduced hormonal response can be found after long-term ECT [107,108], indicating a normalization of HPA axis activity. Along with these findings, there are several studies reporting a long-term decrease in stress hormone cortisol levels following ECT (e.g., [109]).

ECT increased several monoamines after long-term treatment in patients [110]. In addition, animal studies showed that ECT enhanced neurogenesis [111] and neuroplasticity [112], processes which both are known to play important roles in antidepressant-like effects. Neuroimaging studies showed that the hippocampus, the amygdala, prefrontal cortex, anterior cingulate cortex and basal ganglia are main target brain regions of ECT [113]. Amygdala and hippocampus volumes increased after ECT [113], together with an improved functional connectivity [113]. Benson-Martin and colleagues provided an overview of the genetic pathways involved in the mechanisms of action of ECT treatment [114]. Finally, de Jong and colleagues and others reported that ECT resulted in a robust impact on epigenetic mechanisms [115,116].

### 3.2. Clinical Efficacy and the Role of Gender

Interestingly, although ECT is a very old technique, there is not much evidence for gender-specific aspects of ECT treatment reported in literature. A retrospective comparison from 2005 analysed patients with MDD, bipolar disorder and schizophrenia and found that women with MDD and schizophrenia received their first ECT session earlier meaning they had less previous antidepressant drug trials than men [117]. In addition, the authors of this study reported that ECT was more effective in women with schizophrenia compared to male schizophrenic patients [117]. In a study including ECT cases conducted in a hospital in Turkey, the authors did not find a difference between men and women in response to ECT across all tested diagnoses [118]. A study by Bousman et al. reported that the beneficial effect of a catechol-O-methyltransferase polymorphism on ECT response can be exclusively found in male patients [119]. In summary, there is very little knowledge about the role of gender in ECT and more studies are needed to fill this lack of knowledge.

## 4. Conclusions

There is a considerable amount of evidence collected from both animal and human studies, highlighting the central role of gender in depression pathophysiology and treatment. A better understanding of the molecular mechanism activated by hormones both in health and psychiatric disorders combined with a precise knowledge of the pharmacological interaction between hormones and antidepressant drugs, would be suited to redefine treatment guideline and possible identify molecular targets relevant for drug discovery and for gender personalized therapy.

## Figures and Tables

**Figure 1 ijms-20-00949-f001:**
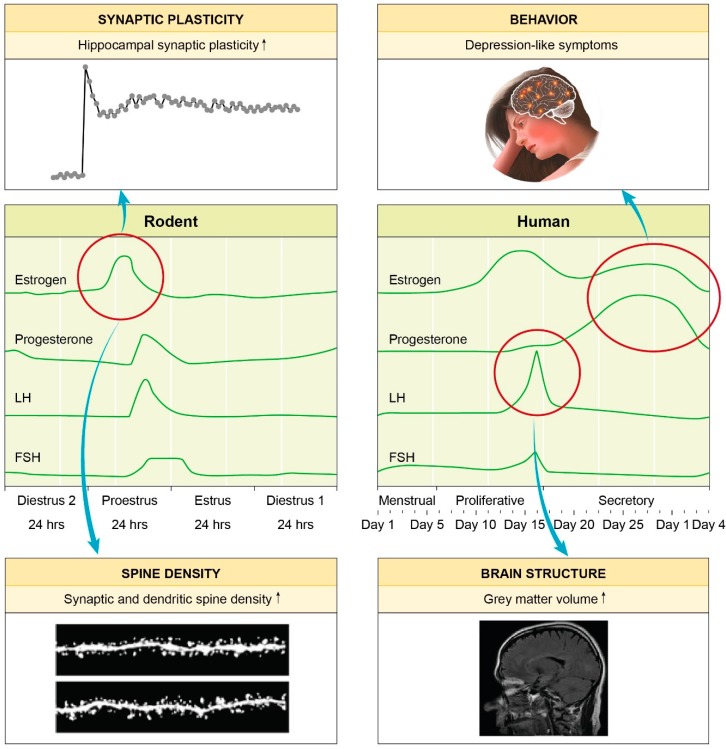
Highlighting the hormonal impact on the female brain: changes throughout the menstrual cycle. Rodents and women share a similar menstrual cycle pattern. Alterations in hormone levels are known to play an important role in neurobiology and mental health. The increased levels of oestrogen in the rodent proestrus phase have been reported to guide an increase in both hippocampal spine density [30] and plasticity [31]. In women, a significant increase in grey matter volume was found around ovulation [32]. The decrease of circulating hormones at the end of the menstrual cycle may induce depression-like symptoms [33].

**Table 1 ijms-20-00949-t001:** In vitro and in vivo experiments using ketamine treatment.

Publication	Test Subject	Study Design	Antidepressant-Like Effect	Molecular Mechanism
Franceschelli et al., 2015 [64]	Male and female C57/BL6J mice	KET in naïve and CMS animals: female and male mice (FST)	KET effect: Female mice > male mice	Effects on excitatory amino acids (glutamate and aspartate), serotoninergic activity.
Saland et al., 2018 [65]	Male and female Sprague-Dawley rats	KET metabolism and distribution		↑ level of KET and NK in both brain and plasma
Ho et al., 2018 [66]	Human iPSC-derived astrocytes	Oestrogen + KET in vitro	Oestrogens augmented the effect of KET	↑ level of AMPA receptor subunit and ERα. Oestrogens: ↑ level of CYP2A6 and CYP2B6.
Dossat et al., 2018 [67]	Male and female C57/BL6J mice	Oestrogen and Progesterone receptor agonist and KET (FST)	Female in proestrus + KET: sensitive to lower dose.	Proestrus female ↑ p-Akt and p-CaMKIIα.
Sarkar et al. 2016 [28]	Male and female Sprague-Dawley rats	KET and social isolation stress (behaviour and synaptic protein level)	IS: male depression like behaviour at 8 weeks while female at 11 weeks. KET rescued the phenotype.	Decline in spine density and synaptic proteins reversed by KET only in male but not female

We list relevant, ketamine-associated publications with significant impact in the field. *KET* ketamine, *CMS* chronic mild stress, *FST* forced swim test, *NK* norketamine, *iPSC* induced Pluripotent Stem Cells, *AMPA* α-amino-3-hydroxy-5-methyl-4-isoxazolepropionic acid receptor, *CYP* Cytochrom P 450 Enzyme, *IS* Isolation Stress.

**Table 2 ijms-20-00949-t002:** In vitro and in vivo experiments using HNK treatment.

Publication	Test Subject	Study Design	Antidepressant-Like Effect	Molecular Mechanism
Zanos et al. 2016 [69]	Male and female C57/BL6J mice	KET: female and male mice (FST) Ketamine metabolites: male mice (CSD, FST, ST)	KET: Female mice > male mice HNK: HNK > (2S,6S)-hydroxynorketamine HNK: lacks ketamine-related side effects	HNK-effects independent of NMDAR-signalling by AMPAR-signalling
Yamaguchi et al. 2018 [70]	Male C57/BL6 mice	LPS with (R)-ketamine and HNK (FST, TST)	(R)-ketamine > HNK Blocking CYP: (R)-ketamine effects ↑	(R)-ketamine and HNK in plasma, brain, CSF
Chou et al. 2018 [72]	Male and female Sprague-Dawley rats	LH with ketamine metabolites (FST, SPT)	HNK: Male ≈ Female (2S,6S)-hydroxynorketamine: no effect	HNK: enhancement of AMPAR-signalling in vlPAG
Pham et al. 2018 [73]	Male BALB/cJ mice	Local (mPFC) and systemic injection of KET and HNK (FST)	Local injection: HNK ≈ KET Systemic injection: HNK ≈ KET	HNK+KET: extracellular 5-hydroxytryptamine (mPFC) ↑, extracellular glutamate (mPFC) ↑ KET: extracellular GABA ↑
Cavalleri et al. 2018 [74]	Murine and human DA neurons	KET and HNK in vitro	-	HNK+KET: structural plasticity ↑ (arborization ↑, soma size ↑)
Collo et al. 2018 [75]	Human DA neurons (PSCs)	KET and HNK in vitro	-	HNK+KET: structural plasticity ↑ (dendrite length ↑ and number ↑)
Yao et al. 2017 [76]	Male C57/BL6 mice	KET and HNK tested ex vivo with electrophysiology	-	HNK+KET: lasting modulation of AMPAR and synaptic plasticity (NAc+VTA), potentiation ↓ and depression ↑ of GA synapses (NAc+VTA-DA neurons)
Shirayama et al. 2018 [77]	Male Sprague-Dawley rats	LH with ketamine metabolites (CAT)	KET: antidepressant-like effect HNK: no effect	-
Yang et al. 2017 [78]	Male C57/BL6 mice	LH and CSD with KET and HNK (FST, TST, SPT)	KET: antidepressant-like effect HNK: no effect	-

We list relevant, HNK-associated publications with significant impact in the field. *HNK* (2R,6R)-hydroxynorketamine, *KET* (R,S)-ketamine, *FST* Forced Swim Test, *CSD* Chronic Social Defeat Model of Depression, *ST* Sociability Test, *NMDAR* N-methyl-D-aspartate receptor, *AMPAR* α-amino-3-hydroxy-5-methyl-4-isoxazolepropionic acid receptor, *LPS* Lipopolysaccharide Model of Depression, *TST* Tail Suspension Test, *CYP* Cytochrom P 450 Enzyme, *CSF* Cerebrospinal Fluid, *LH* Learned Helplessness Model of Depression, *SPT* Sucrose Preference Test, *vlPAG* ventrolateral Periaqueductal grey, *mPFC* medial Prefrontal Cortex, *GABA* gamma-aminobutyric acid, *DA* dopaminergic, *PSC* pluripotent stem cell, *NAc* Nucleus Accumbens, *VTA* Ventral Tegmental Area, *GA* glutamatergic, *CAT* Conditioned Avoidance Test.

**Table 3 ijms-20-00949-t003:** Other rapid-acting antidepressant agents.

Agent	Molecular Target	Reference
Scopolamine	M1/2-antagonist, AMPAR↑, mTOR↑	[81,82]
GLYX-13	Partial agonist and modulator of NMDAR, AMPAR↑	[83,84]
MGS0039, LY3020371	mGlu2/3 antagonists, AMPAR↑	[85,86,87]
L-655,708, MRK-016	NAM of α5-GABA_A_-R, cortex & HC-specific	[88,89]
Cannabidiol	5-HT_1A_-R↑, CB1↑, vmPFC	[90,91,92,93]
Psychedelics (LSD, DOI, DMT, MDMA)	TrkB→mTOR↑+BDNF↑, 5-HT_2A_-R↑, PFC	[94]

We list several other important compounds, which have been shown to provide antidepressant-like effects similar to ketamine and its metabolites. *M1/2* Muscarinic acetylcholine Receptor Type 1 and 2, *AMPAR* α-amino-3-hydroxy-5-methyl-4-isoxazolepropionic acid receptor, *mTOR* mammalian target of rapamycin, *mGlu2/3* Metabotropic glutamate receptor type 2 and 3, *NAM* negative allosteric modulator, *α5-GABA_A_-R* Alpha 5 subunit of the gamma-aminobutyric acid type A receptor, *HC* hippocampus, *NMDAR* N-methyl-D-aspartate receptor, *5-HT_1A_-R* 5-Hydroxy tryptophan receptor type 1A, *CB1* Cannabinoid receptor type 1, *vmPFC* ventromedial prefrontal cortex, *LSD* lysergic acid diethylamide, *DOI* (2,5)-dimethoxy-4-iodoamphetamine, *DMT* N,N-dimethyltryptamine, *MDMA* (3,4)-methylenedioxymethamphetamine, *TrkB* Tyrosine receptor kinase B, *BDNF* Brain-derived neurotrophic factor, *5-HT_2A_-R* 5-Hydroxy tryptophan receptor type 2A, *PFC* Prefrontal cortex.

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
