# Peer review of "Decoding the Mechanism of Action of Rapid-Acting Antidepressant Treatment Strategies: Does Gender Matter?"

_ijms, 2019, doi:10.3390/ijms20040949_

Reviewer 1 Report

The purpose of the review presented by Herzog and colleagues is to draw attention to the inadequacy of treating MDD in relation to gender. The review is well written with clear presentation of the main argument. I have one minor comment:

Reading the manuscript, I laud the authors for their effort to comprehensively report evidence for and against the role of gender in the treatment of MDD. However, I also believe that the review would have benefited more, if the authors attempted to provide more in depth discussion about this discrepancy. In its current form the review does not attempt to explain the different conclusions reached by the different reports, i.e. some studies claiming a clear sex effect on pharmacological treatment between females and males, while other studies failing to report such effect. I believe such discussion to improve the quality of the review, sparking a greater interest among the medical community in the treatment of MDD.    

Author Response

Q1

The purpose of the review presented by Herzog and colleagues is to draw attention to the inadequacy of treating MDD in relation to gender. The review is well written with clear presentation of the main argument. I have one minor comment: Reading the manuscript, I laud the authors for their effort to comprehensively report evidence for and against the role of gender in the treatment of MDD. However, I also believe that the review would have benefited more, if the authors attempted to provide more in depth discussion about this discrepancy. In its current form the review does not attempt to explain the different conclusions reached by the different reports, i.e. some studies claiming a clear sex effect on pharmacological treatment between females and males, while other studies failing to report such effect. I believe such discussion to improve the quality of the review, sparking a greater interest among the medical community in the treatment of MDD.

Authors´ response:

We appreciate the positive feedback and comment from Reviewer 1. We agree that there are contrary and inconsistent reports available regarding gender differences of antidepressant treatment, therefore we included in this manuscript the review by Sramek et al., which summarizes the evidence concerning gender differences in antidepressant treatment and reports all the relevant meta-analyses. Moreover, we have added a small paragraph to the discussion in section 1.2 in order to further improve the manuscript.

“The inconsistencies within those reports are not easy to be explained. One reason might be highly different inclusion and exclusion criteria of clinical trials. These might lead to different patient stratification, thus creating a huge bias and complicating the interpretation of the results. Another possible reason might be that psychiatric nosology and diagnosing is a highly artificial process. Two persons diagnosed with depression may express completely different symptoms, however in clinical trials, they might be put together as namely “depressed patient”. A recent National Institute of Mental Health (NIMH) initiative called the Research Domain Criteria (RDoC), introduces a transdiagnostic approach suited to specify symptoms and phenotypes at individual level[50]. ”

Reviewer 2 Report

This is an interesting observation by the authors. The major comments I have for the paper is selective choice of the anti-depressants. There are tons of anti-depressants available in the market. But, the authors have picked up only some like ketamine, HNK, ECT and show their gender-biased effects.

I don't really understand the purpose of adding the section 2.3. This does not add any value to the topic of the article presented.

I think the Figure 1 can be better explained in the text. I am also missing a section where the hormonal impact on the female brain is discussed in details.

Author Response

Q1

This is an interesting observation by the authors. The major comments I have for the paper is selective choice of the anti-depressants. There are tons of anti-depressants available in the market. But, the authors have picked up only some like ketamine, HNK, ECT and show their gender-biased effects.

Authors´ response:

We agree with the reviewer that there are a lot of antidepressants available on the market. Classical antidepressants have an effect latency of 2-4 weeks, which represents a severe limit in the treatment of psychiatric disorders. In recent years, antidepressants with a more rapid onset of action have raised interests in both clinical and preclinical research. There are already data available on the molecular aspects shaping gender differences of general (classical) antidepressant treatment (e.g. see review by Sramek et al. or LeGates et al.; publications, which we also mentioned throughout the present manuscript). However, we think that less is known on the gender differences of rapid acting antidepressant drugs. With this review we aimed at summarizing, discussing and challenging the available literature on the gender differences of rapid-acting antidepressants, thus we focused on ketamine, HNK, psychedelic agents, and ECT , which all share a rapid onset of antidepressant action. We are convinced that this is of high interest to both clinical and preclinical scientists and potential readers of the current issue

Q2

I don't really understand the purpose of adding the section 2.3. This does not add any value to the topic of the article presented.

Authors´ response:

We thank the Reviewer for this comment. With the present manuscript we want to discuss the current state-of-the-art regarding rapid-acting antidepressant treatment approaches. The oldest non-pharmacological approach is the ECT. Recently ketamine (and very recently its metabolite HNK) revealed an antidepressant rapid onset of action. Interestingly, several other novel compounds showed rapid onset of action. We agree that - due to limited sample sizes and limited clinical trials with these drugs - there is a lack of information about gender differences in those studies. However, we presented the available literature on these novel antidepressant drugs, that might be of high interest for the developing of new compound with fast-acting antidepressant properties.

Q3

I think the Figure 1 can be better explained in the text. I am also missing a section where the hormonal impact on the female brain is discussed in details.

Authors’ response:

We agree with Reviewer 2 that the content of Figure could be explained in more detail throughout the manuscript. We updated section 1.1 accordingly. A detailed section about the various aspects of the hormonal impact on the female brain was not the main focus of our review article. However, we agree that this part is of high interest to the reader. Taking into account the Reviewer´s comments, we have added a recent review article about the interplay of hormones and the brain across genders.

“This is supported by a plethora of interesting findings (Figure 1): Hormonal alterations during the menstrual cycle affect females across species. Elevated estrogen levels mediate an enhanced hippocampal spine density [30] and plasticity [31]. Hagemann et al. revealed that during ovulation women had a significant increase in grey matter volume [32]. Finally, the end of the menstrual cycle and by that, the decline of sexual hormones estrogen and progesterone may induce depression-like symptoms [33]. Sex differences in the stress-induced remodeling of dendrites and synapses in brain regions such as the hippocampus or the prefrontal cortex first emerge in puberty [34], thus highlighting the link between female susceptibility to stress related disorders and hormone effects, mainly estrogen and progesterone [14]. For review article summarizing the hormonal impact on the female brain see [13].”

Reviewer 3 Report

In the present review, The Authors summarized the evidences supporting the importance of gender in modulating response to rapid acting antidepressant treatment (such as ketamine, ECT etc.). Overall, I found the paper very interesting, well written and scientifically sound: enjoyed reading it! I have only some suggestions aimed to further improve the high quality of the paper:

1) In the introduction, I would suggest Authors add a brief note on limitations of current antidepressant treatments (e.g., loss of efficacy in the long term, manic switches, adverse effects, etc.) with appropriate references (see Fornaro et al. Pharmacol Res.2019 and J Affect Disord 2018)

2) I believe that might be useful for the reader if the Authors add a brief note on how literature was searched and selected.

3) Concerning ketamine, its rapid action on suicide ideation and potential neurobiological underpinnings of this peculiar action should be briefly explained too (see De Berardis et al Int J Mol Sci 2018 and Tomasetti et al. Int J Mol Sci. 2017)

Author Response

In the present review, The Authors summarized the evidences supporting the importance of gender in modulating response to rapid acting antidepressant treatment (such as ketamine, ECT etc.). Overall, I found the paper very interesting, well written and scientifically sound: enjoyed reading it! I have only some suggestions aimed to further improve the high quality of the paper:

Q1

In the introduction, I would suggest Authors add a brief note on limitations of current antidepressant treatments (e.g., loss of efficacy in the long term, manic switches, adverse effects, etc.) with appropriate references (see Fornaro et al. Pharmacol Res.2019 and J Affect Disord 2018).

Authors’ answer:

We thank the Reviewer for this comment. We have added appropriate references in paragraph 1.3

“A possible loss of antidepressant treatment efficacy in long term-treated patients, the danger of manic switches, and several prominent side effects are further limitations of antidepressant treatment [54,55].”

Q2

I believe that might be useful for the reader if the Authors add a brief note on how literature was searched and selected.

Author´s answer:

We thank the Reviewer for this comment. We have added those information int the section “Supplementary Material”.

“The literature research was carried on in Pubmed with the following MeSH terms: “Sex; Sex characteristics; Antidepressive Agents; Electroconvulsive Therapy; Hallucinogens; Ketamine. The following Non-MeSH terms were used: (2R,6R)-hydroxynorketamine; rapid-acting antidepressant; psychedelics. The publications were selected if the publication date was less than 10 years.”

Q3

Concerning ketamine, its rapid action on suicide ideation and potential neurobiological underpinnings of this peculiar action should be briefly explained too (see De Berardis et al Int J Mol Sci 2018 and Tomasetti et al. Int J Mol Sci. 2017)

Authors´answer:

We thank the Reviewer for this comment. We have added the appropriate references in paragraph 2.1.

“Furthermore, its rapid action on suicide ideation significantly improves the management of acute MDD treatment [60,61].”

Round  2

Reviewer 2 Report

The author has fulfilled all the criteria required for the article.